# CHATEVAL: TOWARDS BETTER LLM-BASED EVALUATORS THROUGH MULTI-AGENT DEBATE

**Chi-Min Chan**[1], **Weize Chen**[1], **Yusheng Su**[1], **Jianxuan Yu**[1], **Wei Xue**[2],
**Shanghang Zhang**[3], **Jie Fu**[2], **Zhiyuan Liu**[1]*
[1] Tsinghua University
[2] Hong Kong University of Science and Technology
[3] Peking University
`zorowin123@gmail.com`

## ABSTRACT

Text evaluation has historically posed significant challenges, often demanding substantial labor and time cost. With the emergence of large language models (LLMs), researchers have explored LLMs' potential as alternatives for human evaluation. While these single-agent-based approaches show promise, experimental results suggest that further advancements are needed to bridge the gap between their current effectiveness and human-level evaluation quality. Recognizing that best practices of human evaluation processes often involve multiple human annotators collaborating in the evaluation, we resort to a multi-agent debate framework, moving beyond single-agent prompting strategies. In this paper, we build a multi-agent referee team called **ChatEval** to autonomously discuss and evaluate the quality of different texts. Our experiments on two benchmarks illustrate that ChatEval delivers superior accuracy and correlation in alignment with human assessment. Furthermore, we find that the diverse role prompts (different personas) are essential in the multi-agent debate process; that is, utilizing the same role description in the prompts can lead to a degradation in performance. Our qualitative analysis also shows that ChatEval transcends mere textual scoring, offering a human-mimicking evaluation process for reliable assessments.

## 1 INTRODUCTION

Evaluating the quality of text generated by language models or written by humans has long been a challenging endeavor, consistently garnering substantial attention (Celikyilmaz et al., 2020). Traditional methodologies predominantly rely on human annotation of texts (Callison-Burch, 2009), an approach considered overly demanding in terms of time and cost. Automatic evaluation metrics based on n-grams, such as Rouge (Lin, 2004), BLEU (Papineni et al., 2002), and METEOR (Banerjee & Lavie, 2005), have been proposed to tackle this issue (Kondrak, 2005). However, these methods have been shown to exhibit a relatively weak correlation with human judgments, particularly in the context of tasks involving open-ended generation or requiring domain-specific expertise (Novikova et al., 2017).

In view of the impressive text understanding and instruction-following capabilities of recent LLMs, a body of literature (Liu et al., 2023b; Chiang & Lee, 2023; Gao et al., 2023; Shen et al., 2023) has adopted LLM as an evaluator to assess the quality of responses to open-ended questions or traditional NLG tasks, including dialogue response generation and summarization. This methodology is dubbed LLM-as-a-judge (Zheng et al., 2023). Findings from these researches indicate that LLM can mimic human behavior and provide evaluations that correspond with human judgments, revealing a potentially scalable and transparent alternative to costly and laborious human evaluations.

While a *single* powerful LLM can already tackle various missions, emerging studies suggest that *multiple* LLMs can further improve one another through debate and cooperation (Li et al., 2023a; Liang et al., 2023). By incorporating multiple LLMs into an integrated group and designing specific

---

*Corresponding author. Email: `liuzy@tsinghua.edu.cn`

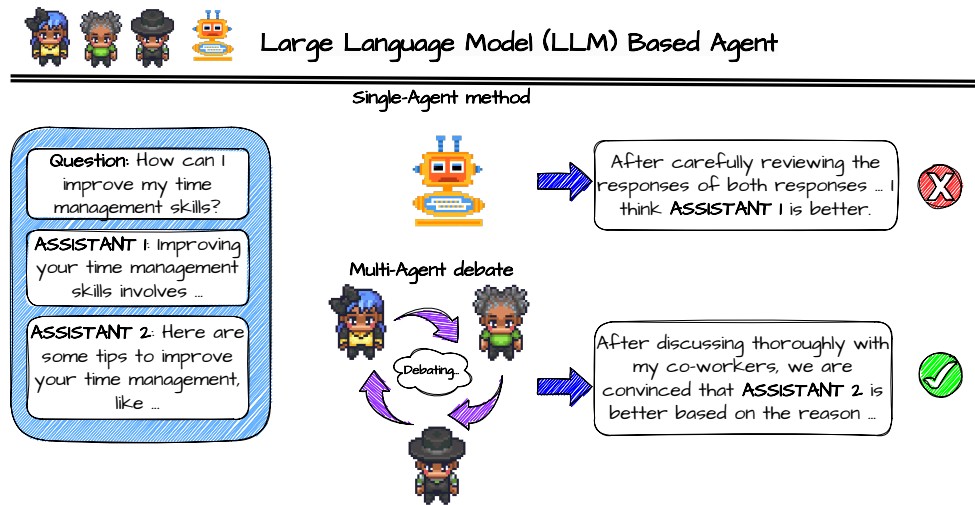

Figure 1: When several referees participate in the evaluation process, they can discuss with each other and finally give a judgment that is better aligned with human annotators.

interaction mechanisms, different LLMs can engage in proposing and deliberating unique responses and thought processes across several rounds. This approach leads to enhanced factuality of generated responses (Du et al., 2023) and improvement in the completion of arduous tasks (Li et al., 2023a; Qian et al., 2023). Furthermore, the multi-agent group also addresses and mitigates the Degeneration-of-Thought (DOT) problem (Liang et al., 2023).

In the human evaluation processes, relying on a single perspective can introduce bias and instability in the results (Karpinska et al., 2021). Recognizing this, best practices often involve multiple human annotators collaborating in the evaluation (Van Der Lee et al., 2019). Drawing inspiration from this collaborative and iterative human evaluation approach, we propose ChatEval, a system that enables each agent to employ varied communication strategies in collaborative discussion, working towards formulating final judgments. Furthermore, to enrich the evaluation dynamics, every agent within ChatEval is endowed with a unique persona. This deliberate design ensures that each agent focuses on distinct perspectives or brings specific expertise to the table. By doing so, the collective evaluation benefits from a more comprehensive lens, capturing nuances and subtleties that a single perspective might overlook. Another underlying intuition of our work stems from renowned concepts in sociology and biology, including *Collective Intelligence*(Woolley et al., 2010) and *Cognitive Synergy*(Luppi et al., 2022), where multiple cognitive processes or systems interact and cooperate in a way that produces a combined effect greater than the sum of their separate effects.

To summarize, the main contribution of our work is as follows:

1. We propose a multi-agent-based framework called **ChatEval** that aligns better with human preferences compared with single-agent-based approaches as depicted in Figure 1.
2. We propose various communication strategies and demonstrate the necessity of diverse role prompts in multi-agent debate scenarios.
3. In the qualitative study, we demonstrate that our agents exhibit human-like behavior, capitalizing on the richness and complexity of language interaction. This elevates ChatEval from being merely a evaluation tool to an embodiment of interactive natural language dialogue.

## 2 METHODOLOGY

In this section, we elaborate on the principal components in ChatEval including *debater agents*, *diverse role specification*, *communication strategy*, and provide a detailed overview of each component's role and functionality.

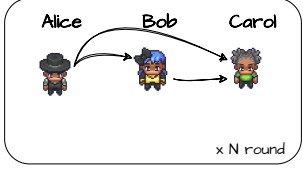 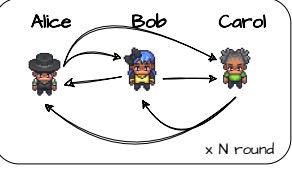 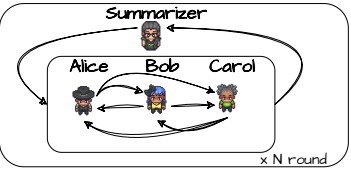

(a) One-by-One   (b) Simultaneous-Talk   (c) Simultaneous-Talk-with-Summarizer

Figure 2: The overall schematic diagram of our proposed three different kinds of communication strategy. The direction of the arrows represents the flow of information, meaning that what this person says will be appended to the chat history of the person pointed to by the arrow.

**Debater Agents**. Debater agents are one of the most significant components in our framework. We treat each individual LLM as an agent and ask them to generate their response from the given prompts[1]. Responses from other agents are served as chat history which will be replaced in the prompt template. After configuring the agents, we then start the group debate where each agent autonomously receives responses from the others and, in turn, delivers its own responses to them. It should be noted that the whole process does not require human intervention.

**Diverse Role Specification**. As presented in Section 1, diverse role specification is necessary for the framework as well. Although all the agents share a common prompt template, we substitute the *role_description* slot with diverse role prompts, specifying distinct personalities for different agents. We take inspiration from Wu et al. (2023) and formulate an analogous role description.

**Communication Strategy**. How to maintain the chat history is another significant issue in ChatEval. In our work, we use a more intuitive term to illustrate the maintenance of the chat history called *communication strategy*. In a nutshell, different communication strategies can be seen as different approaches to maintaining and manipulating their chat history. As is shown in Figure 2, We primarily design three different communication strategies and the full description and algorithm of the above communication strategies can be found in Appendix C.

## 3 EXPERIMENTS

We evaluate ChatEval on two benchmarks, *FairEval* and *Topical-Chat* which represent the categories of open-ended question answer and dialogue response generation, respectively. For the LLMs in ChatEval, we choose to use ChatGPT(GPT-3.5-turbo) and GPT-4 considering their strong capability shown in the past literature, we also test our frame work on smaller open-sourced model such as Llama2-Chat-7B and Vicuna, the results of which are shown in Appendix D, and the detailed settings for the experiments are discussed in Appendix B.

### 3.1 BENCHMARKS

The detailed introduction of different categories and benchmarks are listed as follows:

**Open-ended Question Answer** is a key component within the field of NLP and generative AI. It necessitates an AI system to provide comprehensive, detailed, and human-like responses to questions that don't have a predefined or fixed set of possible answers. The work of Chiang et al. (2023) encompasses a collection of 80 open-ended questions originating from a wide array of categories, including common-sense, counterfactual, coding, etc. We then take the human annotation results from Wu et al. (2023) to conduct the experiments in this paper. For each question, they direct three annotators to evaluate the replies given by Vicuna-13B and ChatGPT through the given rules and finally derive the results by the majority votes among the annotators.

**Dialogue Response Generation** is a task involves creating a coherent and contextually appropriate response to a given input dialogue. We draw upon the *Topical-Chat* (Gopalakrishnan et al., 2019) dataset for our study. We then take the human annotation results from Mehri & Eskenazi (2020) where they carry out the annotations on 60 dialogue contexts with each response generated by 6

---

[1]The full prompt template can be found in Appendix A.

different systems. Human evaluators analyzed these responses based on *natural*, *coherence*, *engagingness*, *groundedness*, and *understandable*, where we take the first four dimensions for experiments in our paper following Zhong et al. (2022).

## 3.2 BASELINES

We evaluate ChatEval against following methods. As the main portion of our comparison, we primarily focuses on the single-agent-based method. **Single-Agent** means that we directly query an LLM to generate the response towards the evaluation[2]. **Multi-Agent** means that we employ multiple LLMs, either in an ensemble or through a debate approach. When we do the ensemble, we apply the single-agent method multiple times using different role prompts across the same instance and then aggregate their results through averaging to derive the final outcome. By default, we configure the communication strategy to one-by-one, agent numbers to 2, and discussion turns to 2 in this section and employ position calibration techniques in both single-agent and multi-agent settings. We will discuss more debate configurations in Section 4 for completeness. For the open-ended question answer task, we also compare our method with a simple ensemble method and **FairEval** (Wang et al., 2023b). They propose various strategies to improve the evaluation performance of a LLM including Multiple Evidence Calibration (MEC) and Balanced Position Calibration (BPC). For the dialogue response generation task, we also compare our method with **G-EVAL** (Liu et al., 2023b). They utilize CoT and probability-weighted summation for their method. Additionally, we include results from n-gram-based metrics, such as **ROUGE** (Lin, 2004), **BLEU** (Papineni et al., 2002) and embedding-based metrics such as **BERTScore** (Zhang et al., 2019).

## 3.3 RESULTS FOR OPEN-ENDED QUESTION ANSWERS

We adopt the same evaluation approach as Wang et al. (2023b) to assess the annotation results produced by different methods and annotators. Specifically, we calculate the Accuracy (Acc.), which measures the proportion of correctly classified instances out of the total instances, and the Kappa correlation coefficient (Kap.) (McHugh, 2012) which gauges the agreement between results from models and human annotators while taking into account the possibility of agreement occurring by chance. Both metrics provide insights into the reliability and consistency of the annotations. We take the human annotation results and FairEval's (Wang et al., 2023b) best results from their paper. As is shown in Table 1, different annotators can reach a relatively high agreement and perform better than any other LLM-based approach. Still, the average human annotations accuracy which is 71.7% shows there exists a certain degree of discrepancy among different unique individuals revealing that text evaluation is absolutely an arduous task. The second part and the third part of Table 1 show the results of FairEval's method and the results of our proposed method respectively. We find that (1) ChatEval can enhance the performance of the evaluation process, achieving higher alignment with human preference compared with single-agent evaluation. Specifically, the multi-agent-based method improves the accuracy by 6.2% for ChatGPT and 2.5% for GPT-4; (2) ChatEval surpasses FairEval's best results within both ChatGPT and GPT-4 settings showing the effectiveness of our proposed method; (3) Compared to ChatEval, a basic ensemble fails to markedly improve the evaluator's performance, highlighting the crucial role of natural language interaction in our framework.

## 3.4 RESULTS FOR DIALOGUE RESPONSE GENERATION

For the dialogue response generation benchmarks, we align the evaluation method with Zhong et al. (2022), calculating the turn-level Spearman and Kendall-Tau correlation in correspondence with human judgments on four aspects (*naturalness*, *coherence*, *engagingness* and *groundedness*). Results can be found in Table 2. In the first part of Table 2, we demonstrate that n-gram-based metrics and embedding-based metrics perform overall poorly on all the aspects evaluated illustrating that these methods can hardly reveal human preference. In the second part of Table 2, we show the results from the G-eval (Liu et al., 2023b) paper. They first ask the LLM to generate intermediate thought and finally calculate the weighted summation of the output scores based on the probability. The results show that their method outperforms previous traditional metrics depicting the fact

---

[2]We use the same prompt template as our multi-agent debate settings in single-agent baseline except that we ignore some slot.

Table 1: Accuracy (Acc.) and Kappa correlation coefficient (Kap.) of different methods on FairEval. We present our results with average and standard deviation by running the experiment five times.

| Evaluator | Methods | Acc. (%) | Kap. |
|---|---|---|---|
| **Human** | | | |
| Annotator1 | - | 68.8 | 0.5 |
| Annotator2 | - | 76.3 | 0.62 |
| Annotator3 | - | 70 | 0.5 |
| **FairEval** | | | |
| ChatGPT | MEC+BPC | 58.7 | 0.31 |
| GPT-4 | MEC+BPC | 62.5 | 0.37 |
| **Ours** | | | |
| ChatGPT | Single-Agent | $53.7_{\pm 1.4}$ | $0.27_{\pm 0.02}$ |
| ChatGPT | Multi-Agent (Ensemble) | $55.5_{\pm 0.7}$ | $0.29_{\pm 0.01}$ |
| ChatGPT | Multi-Agent (ChatEval) | $\mathbf{60.0}_{\pm 0.9}$ | $\mathbf{0.30}_{\pm 0.02}$ |
| GPT-4 | Single-Agent | $60.8_{\pm 0.7}$ | $0.36_{\pm 0.01}$ |
| GPT-4 | Multi-Agent (Ensemble) | $61.5_{\pm 0.5}$ | $0.38_{\pm 0.01}$ |
| GPT-4 | Multi-Agent (ChatEval) | $\mathbf{63.8}_{\pm 0.9}$ | $\mathbf{0.40}_{\pm 0.01}$ |

Table 2: Turn-level Spearman ($\rho$) and Kendall-Tau ($\tau$) correlations of different methods on Topical-Chat benchmark, **SA** means Single-Agent, **EN** means Multi-Agent (Ensemble) and **MA** means Multi-Agent (ChatEval). Our ChatGPT settings should be compared to G-EVAL-3.5, and GPT-4 settings should be compared to G-EVAL-4.

| Metrics | Naturalness | | Coherence | | Engagingness | | Groundedness | | Average | |
|---|---|---|---|---|---|---|---|---|---|---|
| | $\rho$ | $\tau$ | $\rho$ | $\tau$ | $\rho$ | $\tau$ | $\rho$ | $\tau$ | $\rho$ | $\tau$ |
| ROUGE-L | 0.146 | 0.176 | 0.203 | 0.193 | 0.300 | 0.295 | 0.327 | 0.310 | 0.244 | 0.244 |
| BLEU-4 | 0.175 | 0.180 | 0.235 | 0.131 | 0.316 | 0.232 | 0.310 | 0.213 | 0.259 | 0.189 |
| BERTScore | 0.209 | 0.226 | 0.233 | 0.214 | 0.335 | 0.317 | 0.317 | 0.291 | 0.274 | 0.262 |
| G-EVAL-3.5 | 0.539 | 0.532 | 0.544 | 0.519 | 0.691 | 0.660 | 0.567 | 0.586 | 0.585 | 0.574 |
| G-EVAL-4 | 0.565 | 0.549 | 0.605 | **0.594** | 0.631 | 0.627 | 0.551 | 0.531 | 0.588 | 0.575 |
| ChatGPT(SA) | 0.474 | 0.421 | 0.527 | 0.482 | 0.599 | 0.549 | 0.576 | 0.558 | 0.544 | 0.503 |
| ChatGPT(EN) | 0.421 | 0.359 | 0.486 | 0.442 | 0.611 | 0.551 | 0.661 | 0.628 | 0.545 | 0.495 |
| ChatGPT(MA) | 0.441 | 0.396 | 0.500 | 0.454 | 0.664 | 0.607 | 0.602 | 0.583 | 0.552 | 0.510 |
| GPT-4(SA) | 0.532 | 0.483 | 0.591 | 0.535 | 0.734 | 0.676 | **0.774** | **0.750** | 0.658 | 0.611 |
| GPT-4(EN) | 0.512 | 0.450 | 0.607 | 0.544 | 0.755 | 0.693 | 0.781 | 0.756 | 0.664 | 0.611 |
| GPT-4(MA) | **0.630** | **0.571** | **0.619** | 0.561 | **0.765** | **0.695** | 0.722 | 0.700 | **0.684** | **0.632** |

that the LLM-based evaluator is effective and reliable for evaluating the dialogue response generation task. While their method delivers sound results, our proposed approach raises the bar in terms of performance for GPT-4. Specifically, ChatEval improves the average Spearman and Kendall-Tau correlation by 0.096 (16.3%) and 0.057 (10.0%) respectively. Additionally, compared with the single-agent method, ChatEval amplifies the performance both for ChatGPT and GPT-4, showing the effectiveness of our method which is aligned with the results in Section 3.3.

## 4 ANALYSIS

In this section, we further explore the key components encompassed in ChatEval. We discuss the importance of diverse role prompts in Section 4.1, the effect of different communication strategies in Section 4.2, the impact of role numbers and discussion turns in Section 4.3 and carry out the qualitative study in Section 4.4. If not specified otherwise, we choose the FairEval benchmark and ChatGPT as the backbone LLM for the analysis.

### 4.1 THE IMPORTANCE OF DIVERSE ROLE PROMPTS

Previously in Table 1 and 2, we demonstrate that ChatEval equipped with diverse role configurations can significantly improve the performance of evaluation. We further consider whether it is necessary to design diverse role prompts for the evaluation system. To answer so, we carry out the experiments by replacing all the role prompts with "*You are now an Annotator, one of the referees in the text evaluation task.*" and keeping other prompts unchanged. We experiment with the one-by-one communication strategy and 2 agents with 2 discussion turns. The results in Table 3 illustrate that ChatEval with the same role prompt design underperforms that with diverse role prompt design and cannot effectively enhance the performance compared with single-agent setting, highlighting the cruciality of diverse role prompt design in the multi-agent debate framework.

Based on the findings above, we are convinced that diverse role prompts are crucial for our framework. Furthermore, we delved deeper to study the effects of different portraits assigned to agents. To analyze this, we referred to the categories proposed by Wang et al. (2023b). We designed specific roles for different groups and compared their evaluation quality with our default setting[3]. We selected four categories from which we could clearly derive specific roles for this experiment. Specifically, the four categories we chose are: generic, coding, writing, and knowledge. As a simple example, when we design the coding group, we recruit experts like *Guido van Rossum* by specifying *"You are Guido van Rossum. You are the creator of the Python programming language. [...]"* in the role prompts. By designating different roles and backgrounds in the role prompts, we can assemble referee teams with specific expertise in various domains.

As shown in Figure 3, our specially designed knowledge, writing, and coding groups can outperform or match the evaluations of other groups in corresponding categories. As for the generic group, we found that it performs well overall compared to its counterparts. The results further underscore the effectiveness of the role prompts and reveal potential avenues for further optimizing the framework, such as using mechanisms like dynamic role prompt specification.

Table 3: Effect of diverse role specification on FairEval benchmark.

| Evaluator | Methods | Acc. (%) | Kap. |
|---|---|---|---|
| ChatGPT | Single-Agent | 53.8 | 0.27 |
| ChatGPT | Multi-Agent (without Diverse Role Prompts) | 53.8 | 0.25 |
| ChatGPT | Multi-Agent (with Diverse Role Prompts) | 60 | 0.33 |

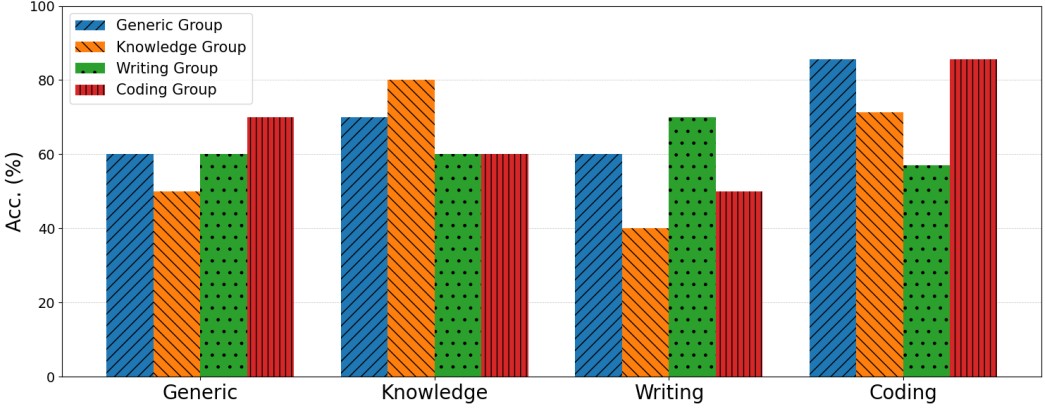

Figure 3: Evaluation quality of different groups on different categories.

---

[3] which includes three different roles: general public, news author, and critic

## 4.2 THE STUDY OF COMMUNICATION STRATEGIES

As shown in Figure 2, we also design three different communication strategy termed as *one-by-one*, *simultaneous-talk*, *simultaneous-talk-with-summarizer*. The detailed descriptions and formal formulations can be found in Appendix C. As depicted in Figure 4, distinct communication strategies exhibit varying behaviors depending on the role-turn configuration. However, they peak at a similar performance range of 60.0%-62.5% in accuracy. Furthermore, the *simultaneous-talk-with-summarizer* strategy appears to offer slightly better scalability, as evidenced by its consistent upward trajectory with increasing role numbers and discussion turns. It is hypothesized that this is because the summarization-style history doesn't expand rapidly with the context length, thereby preserving the reasoning capability of the LLMs. Meanwhile, variations in performance among three different communication strategies underscore the influence of different strategies on the effectiveness of the evaluation quality, revealing the potential for further exploration and optimization of ChatEval. Thus, future studies could be aimed at a more comprehensive understanding of different communication strategies, and how they could be effectively employed to enhance performance. This could serve as an avenue for substantial improvements and novel insights in the multi-agent debate framework.

## 4.3 THE IMPACT OF ROLE NUMBERS AND DISCUSSION TURNS

We then study the impact of different role numbers and discussion turns. From Figure 4a, 4b and 4c, a discernible trend is observed in the relationship between the role number and both Acc. and Kap. As the role number increases, there is a corresponding growth in performance, underscoring the effectiveness of incorporating diverse roles.

Conversely, no significant upward trend is detected with respect to the increase in discussion turns, as is shown in Figure 4d and 4e. This observation aligns with the findings in Liang et al. (2023); Du et al. (2023), highlighting a consistent phenomenon where continual discussion often leads to stagnation or even degradation of performance. As we mentioned before in Section 4.2, such a trend may be attributed to issues associated with the rapidly ever-increasing context length, which consequently diminishes the performance. These results prompt a more nuanced understanding of the balance needed between role-turn dynamics to optimize the performance of ChatEval.

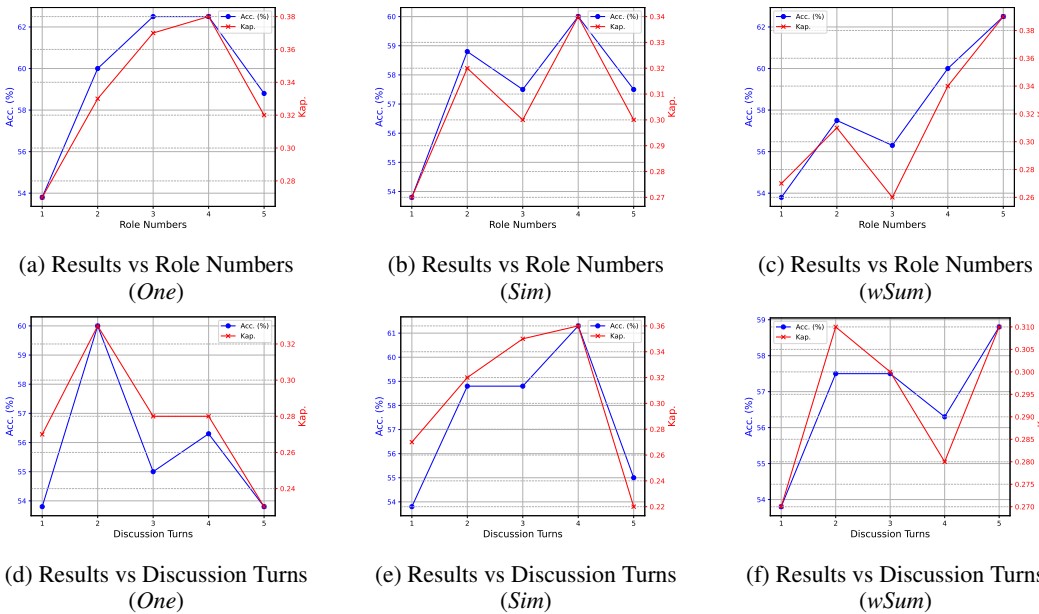

(a) Results vs Role Numbers *(One)*

(b) Results vs Role Numbers *(Sim)*

(c) Results vs Role Numbers *(wSum)*

(d) Results vs Discussion Turns *(One)*

(e) Results vs Discussion Turns *(Sim)*

(f) Results vs Discussion Turns *(wSum)*

Figure 4: Comparison of different configuration of ChatEval on FairEval Benchmark. We use *One*, *Sim* and *wSum* to denote *one-by-one*, *simultaneously-talk* and *simultaneously-talk-with-summarizer* respectively.

## 4.4 QUALITATIVE ANALYSIS

Figure 5 showcases the debate process towards the evaluation of two assistants' responses to the open-ended question "*What are the most effective ways to deal with stress?*". We can find that both of the responses produce similar strategies and equally compelling descriptions for dealing with stress, making it challenging to discern significant disparity in terms of quality. It is in this context of nuanced evaluation that the significance of the ChatEval process emerges.

We can pinpoint several human-like behaviors exhibited by the agents that can enrich our comprehension of the evaluation process; (1) **Opening Statement**: Alice initiates the debate with a clear stance, establishing the foundational argument and guiding the trajectory of the subsequent discourse. (2) **Alternative Proposal**: Bob introduces an alternative viewpoint, emphasizing the need to consider diverse interpretations. This not only broadens the discussion but also stimulates critical thinking. In the context of a debate, the introduction of an alternative proposal prevents the stagnation of thought, challenges pre-existing bias, and uncovers considerations that might otherwise be overlooked, ensuring that the discussions are well-rounded. (3) **Stance Maintenance**: Alice's persistent adherence to her initial stance, even when faced with opposing views, exemplifies commitment and challenges other participants to refine their perspectives. This behaviour ensures the conversation remains robust, focused, and continually evolving, driving all participants to a higher level of engagement and critical thinking. (4) **Seeking Consensus**: The discussion's climax reveals a collective agreement amongst the participants, which is reached through mutual understanding and compromise, underlining the value of each presented viewpoint.

In light of the above, ChatEval stands out not just as a tool for comparison but as an embodiment of interactive natural language dialogue. By simulating human argumentative interactions, it differentiates itself from static, single-presented opinions. This dynamic interaction showcases the richness and complexity of language, capturing nuances often missed in singular viewpoints. As such, ChatEval offers a reliable evaluation process that not only mirrors human discourse but also highlights the transformative power of collaborative dialogue. This positions it uniquely, underscoring its significant potential to execute text evaluation tasks both reliably and effectively.

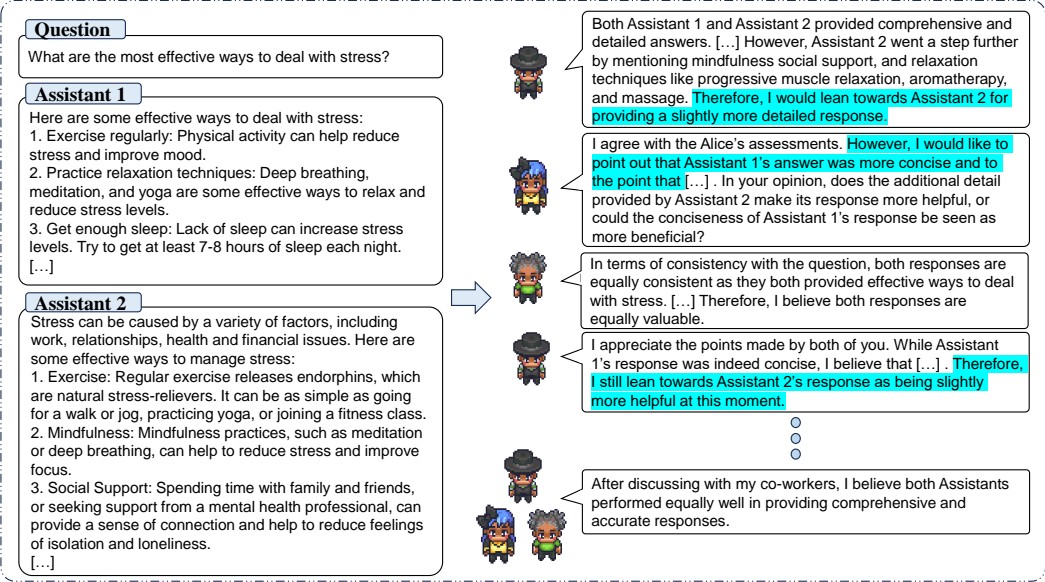

Figure 5: An illustrative example from ChatEval presenting a discussion process involving participants Alice ( ), Bob ( ) and Carol ( ). They initiate a group talk with the purpose of determining the superior response from two assistants. Due to spatial constraints within the layout, we use [...] to omit some redundant text.

## 5 RELATED WORK

**Automatic NLG evaluation** In the landscape of NLG, evaluating the quality of texts represents a particularly arduous task. For a significant period, evaluation was primarily dependent on human annotations that is labor-intensive and limited by scalability issues. Automatic NLG evaluation attempts to address these challenges by leveraging computational models to assess the quality of texts. Previous work lies on the following categories: (1) *n-gram-based metrics*: ROUGE (Lin, 2004) is a set of metrics that compute the amount of overlap between n-grams in the machine-generated summaries and the reference summaries. BLEU (Papineni et al., 2002) compare the generated texts with reference translations, based on the co-occurrence of n-grams in both texts. In spite of being easily and widely used, the above method is incapable of capturing syntactic and semantic similarity (Stent et al., 2005). (2) *embedding-based metrics*: Word embeddings are vector representations of words that capture their semantic properties. A bunch of work leverages word embeddings to evaluate the semantic similarity between two pieces of texts. BERTScore (Zhang et al., 2019) use contextualized word embeddings from transformer models like BERT (Devlin et al., 2018), BLEURT (Sellam et al., 2020) utilize supervised training data to enhance the performance. MoverScore (Zhao et al., 2019) combine contextualized word embeddings with Earth Mover's Distance (Rubner et al., 2000). (3) *LLM-based metrics*: Amidst the flourishing advancement of LLM which embodies a wealth of information derived from extensive training data, using LLM as an evaluator has experienced notable progress. GPTScore (Fu et al., 2023) utilize conditional probability to assign the texts a score representing its quality. Wang et al. (2023a) explore the potential of utilizing ChatGPT as an NLG evaluator by prompting it to score texts directly. Wang et al. (2023c) curate a reliable dataset containing pairwise comparison and evaluation explanation which can be used to train a foundation model making it a better evaluator. Bai et al. (2023) propose decentralized evaluation to provide fairer evaluation results. G-EVAL (Liu et al., 2023b) propose probability-weighted techniques to calibrate the score given by a single LLM.

**Communicative Agents** Most recently, significant attention has been dedicated to the development of communicative agents. These agents, often acted by LLMs like ChatGPT or GPT-4, are designed to interact and communicate effectively with other agents or human users using natural language. The primary goal is to facilitate more productive and efficient interaction and collaboration as different agents can autonomously communicate and negotiate to tackle a more complex task collectively. Several studies have explored various aspects of communicative agents. Li et al. (2023a); Chen et al. (2023) propose a cooperative agent framework enabling agents to autonomously cooperate to solve complex tasks. Park et al. (2023) create a sandbox environment consisting of 25 individual virtual entities endowed with a character description and memory system. Every intelligent agent is capable of autonomously interacting with other agents and the environment simulating reliable human behavior. Dong et al. (2023); Qian et al. (2023) incorporate a waterfall model to manage a multi-agent virtual team towards software development. Liu et al. (2023a) utilize a sandbox environment to curate reliable datasets in better alignment with human preference and train a socially-aligned LLM. Liang et al. (2023) and Du et al. (2023) also make use of the multi-agent debate framework in other scenarios such as translation and arithmetic problems resulting in better results. Wang et al. (2023d) propose an alternative method called self-collaboration to enable the communication of agents by utilizing a single LLM prompted by multi-persona descriptions. Mandi et al. (2023) propose a novel framework designed for the collaboration of multiple robots, utilizing multiple LLMs to enhance coordination and strategic planning among the robots. Concurrent with our work, Li et al. (2023b); Zhang et al. (2023) also propose a similar approach. However, they probe different dimensions of improving LLM-based evaluators and do not explore the effectiveness of natural language interaction.

## 6 CONCLUSION

In this paper, we present evidence that ChatEval contributes to improving the evaluation performance concerning text quality, aligning more closely with human preferences. We emphasize the necessity of the diverse role specification and propose distinct communication strategies as integral components within ChatEval. Our qualitative analysis of the discussion process conveys insightful intuitions about how texts are evaluated by ChatEval and substantiates our approach's ability to support comprehensive evaluations akin to human judgment, thereby demonstrating the reliability and efficacy of our framework.

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

## A  PROMPT TEMPLATE AND DIVERSE ROLE PROMPTS

The overall prompt template is shown in Table 4, we draw inspiration from Wu et al. (2023) and design several different role descriptions as follows.

**General Public** *You are now General Public, one of the referees in this task. You are interested in the story and looking for updates on the investigation. Please think critically by yourself and note that it's your responsibility to choose one of which is the better first.*

**Critic** *You are now Critic, one of the referees in this task. You will check fluent writing, clear sentences, and good wording in summary writing. Your job is to question others judgment to make sure their judgment is well-considered and offer an alternative solution if two responses are at the same level.*

**News Author** *You are News Author, one of the referees in this task. You will focus on the consistency with the original article. Please help other people to determine which response is the better one.*

**Psychologist** *You are Psychologist, one of the referees in this task. You will study human behavior and mental processes in order to understand and explain human behavior. Please help other people to determine which response is the better one.*

**Scientist** *You are Scientist, one of the referees in this task. You are a professional engaged in systematic study who possesses a strong background in the scientific method, critical thinking, and problem-solving abilities. Please help other people to determine which response is the better one.*

## B  IMPLEMENTATION DETAILS

We choose to utilize models from OpenAI's GPT family as our LLMs in ChatEval, including GPT-4 and ChatGPT (GPT-3.5-turbo) and set the temperature to 0 to ensure reproducibility. The rationale

> [Question]
> {source_text}
> [The Start of Assistant 1's Answer]
> {compared_text_one}
> [The End of Assistant 1's Answer]
> [The Start of Assistant 2's Answer]
> {compared_text_two}
> [The End of Assistant 2's Answer]
> [System]
> We would like to request your feedback on the performance of two AI assistants in response to the user question displayed above.
> Please consider the helpfulness, relevance, accuracy, and level of detail of their responses. Each assistant receives an overall score on a scale of 1 to 10, where a higher score indicates better overall performance.
> There are a few other referees assigned the same task, it's your responsibility to discuss with them and think critically before you make your final judgment.
> Here is your discussion history:
> {chat_history}
> {role_description}
> Now it's your time to talk, please make your talk short and clear, {agent_name} !

Table 4: The prompt template for FairEval Dataset. We replace the colored slot with real text before querying the LLMs. Note that we use the same template when conducting single-agent-based experiments and ignore the chat history and role description slot.

behind this selection is the exceptional performance these models offer, being among the most advanced and powerful in the world. Additionally, their accessibility and ease of use through APIs enable us to directly call and interact with the models during our research, significantly simplifying the process. In our current research, we focus on homogeneous groups of LLMs. That is, within a given multi-agent group, all LLMs belong to the same GPT family model, either all GPT-4 or all ChatGPT. We acknowledge the potential of heterogeneous groups for future research, which could provide fascinating insights into how strong models and weak models can cooperate in a multi-agent setting. Additionally, unlike previous work like Du et al. (2023), we do not explicitly ask the debater agents to reach a consensus at the end of the debate. In situations where the response format relies on direct comparison, we derive the final results from the **majority vote** among various annotators. Conversely, if the response format requires a direct score, we calculate the **average** score obtained from multiple annotators. This methodological approach ensures the impartiality and balance of our evaluation process.

## C   FORMAL DEPICTION OF DIFFERENT COMMUNICATION STRATEGY

In this paper, we predominantly adopt the following three different communication strategies, which are:

1. **One-By-One**. During each round of the debate, the debater agents take turns in a set order to generate their response based on the current observation. When it's time for a debater agent to respond, we directly concatenate what previous other agents have said into its chat history slot. Please refer to Algorithm 1.

2. **Simultaneous-Talk**. Unlike the one-by-one strategy, we carry out an alternative communication strategy called simultaneous-talk, where debater agents are prompted to asynchronously generate responses in each iteration of the discussion to nullify the impact of the speaking order. Please refer to Algorithm 2.

3. **Simultaneous-Talk-with-Summarizer**. The main difference between this strategy and simultaneous-talk is that we additionally employ another LLM as a summarizer. At the end of each iteration of the debate, we prompt this extra LLM to summarize the messages

conveyed so far and we replace the chat history slots of all debater agents with this summarization. Please refer to Algorithm 3.

---

**Algorithm 1:** One-by-One

---

**input** : agents number $N$, discuss turn $T$, a group of debate agents $[D_1, \cdots, D_N]$, chat history of each agent $[H_1, \cdots, H_N]$, answer_extracter (either majority vote or average score) $EXT$

**output:** Final results for text evaluation $ANS$

---

1 **for** $t \leftarrow 0$ **to** $T$ **do**
2    **for** $n \leftarrow 1$ **to** $N$ **do**
3       $h_n \leftarrow D_n(H_n)$;
      // utilize agents to generate responses
4       **for** $m \leftarrow n$ **to** $N$ **do**
5          **if** $m > 1$ **then**
6             $H_m \leftarrow H_m + h_n$;
            // concatenate current response to later agents' chat history
7          **end**
8       **end**
9    **end**
10 **end**
11 $ANS \leftarrow EXT([H_1, \cdots, H_N])$;
12 **return** $ANS$;

---

**Algorithm 2:** Simultaneous-Talk

---

**input** : agents number $N$, discuss turn $T$, a group of debate agents $[D_1, \cdots, D_N]$, chat history of each agent $[H_1, \cdots, H_N]$, answer_extracter (either majority vote or average score) $EXT$, buffer $BUF$

**output:** Final results for text evaluation $ANS$

---

1 **for** $t \leftarrow 0$ **to** $T$ **do**
2    **for** $n \leftarrow 1$ **to** $N$ **do**
3       $h_n \leftarrow D_n(H_n)$;
      // utilize agents to generate responses
4       $buf \leftarrow buf + h_n$;
      // add the responses in current turn to the buffer
5    **end**
6    **for** $n \leftarrow 1$ **to** $N$ **do**
7       $H_n \leftarrow H_n + buf$;
      // add the buffer to all agents' chat history
8    **end**
9 **end**
10 $ANS \leftarrow EXT([H_1, \cdots, H_N])$;
11 **return** $ANS$;

---

## D   GENERALIZATION TO SMALLER MODELS

Our primary focus in the paper is on the framework implemented with the most powerful LLMs such as ChatGPT and GPT-4. However, we also carry out the experiments on Llama2-Chat-7b and Vicuna-7b-v1.5. As indicated in the Table 5, Llama2-Chat achieves only marginal performance under single-agent CoT methods. Despite reaching an accuracy of 37.5%, the negative kappa coefficient suggests that the observed agreement is less than what would be expected by chance. However, it is noteworthy that through the application of a multi-agent debate, Llama2-Chat demonstrates a modest improvement, albeit still marginally above chance levels.

---

**Algorithm 3:** Simultaneous-Talk-with-Summarizer

---

**input** : agents number $N$, discuss turn $T$, a group of debate agents $[D_1, \cdots, D_N]$, chat history of each agent $[H_1, \cdots, H_N]$, answer_extracter (either majority vote or average score) $EXT$, buffer $BUF$, summarizer $SUM$

**output:** Final results for text evaluation $ANS$

---

**1 for** $t \leftarrow 0$ **to** $T$ **do**
**2**      **for** $n \leftarrow 1$ **to** $N$ **do**
**3**          $h_n \leftarrow D_n(H_n)$;
         `// utilize agents to generate responses`
**4**          $buf \leftarrow buf + h_n$;
         `// add the responses in current turn to the buffer`
**5**      **end**
**6**      **for** $n \leftarrow 1$ **to** $N$ **do**
**7**          $H_n \leftarrow H_n + SUM(BUF)$;
         `// add the summarized buffer to all agents' chat history`
**8**      **end**
**9 end**
**10** $ANS \leftarrow EXT([H_1, \cdots, H_N])$;
**11 return** $ANS$;

---

In contrast, Vicuna exhibits more robust performance improvement compared to Llama2-Chat. ChatEval notably enhances its capabilities, achieving an accuracy of 52.3% and a kappa coefficient of 0.19, indicative of fair agreement beyond chance. While both models show limitations in their performance, these results demonstrate the effectiveness of ChatEval and underscore its positive impact.

Table 5: Accuracy (Acc.) and Kappa correlation coefficient (Kap.) of smaller models on FairEval.

| Evaluator | Acc. (%) | Kap. |
|---|---|---|
| Llama2-Chat-7B (SA) | $37.5_{\pm 1.3}$ | $-0.01_{\pm 0.01}$ |
| Llama2-Chat-7B (MA) | $40.0_{\pm 1.1}$ | $0.05_{\pm 0.01}$ |
| Vicuna-7B-v1.5 (SA) | $45.0_{\pm 1.5}$ | $0.12_{\pm 0.01}$ |
| Vicuna-7B-v1.5 (MA) | $52.3_{\pm 1.3}$ | $0.19_{\pm 0.01}$ |

Table 6: Average Cost on FairEval.

| Evaluator | Cost | Time |
|---|---|---|
| Human | $90 | 240min |
| GPT-4(SA) | $2.90 | 10min |
| GPT-4(EN) | $8.70 | 10min |
| GPT-4(MA) | $12.30 | 36min |
| ChatGPT(SA) | $0.11 | 3min |
| ChatGPT(EN) | $0.33 | 3min |
| ChatGPT(MA) | $0.41 | 11min |
| G-EVAL-3.5 | $2.20 | 3min |
| G-EVAL-4 | $58 | 10min |
| FairEval-ChatGPT | $0.34 | 3min |
| FairEval-GPT-4 | $6.38 | 10min |

## E   AVERAGE COST

We release the average cost of our framework and other methods against with employing human evaluators. In the Table 6, it is evident that the evaluator based on LLMs significantly reduces both time and financial expenditures. Although the ChatEval framework incurs higher costs than the SA and MA ensemble methods due to multiple inference rounds necessary for final judgment derivation, it remains more cost-effective than employing human evaluators. We consider this an acceptable trade-off for the benefits provided.

