# OpenReview forum: "ChatEval: Towards Better LLM-based Evaluators through Multi-Agent Debate"
_ICLR.cc/2024/Conference — ICLR 2024 poster_

### Official Review · Reviewer_XZLb · 2023-10-31

**Soundness:** 2 fair
**Presentation:** 1 poor
**Contribution:** 2 fair
**Rating:** 5
**Confidence:** 3

**Summary:**

This paper presents ChatEval, a multi-agent debate framework that utilizes multiple LLMs to debate and evaluate the quality of machine-generated texts and improve the quality of the final text as a result of the process. The experiments consider two benchmarks: (1) FairEval along with human annotation results from Wu et al. (2023) – containing 80 open-ended questions with three annotators’ annotations on two model outputs and (2) Topical-Chat along with human annotation results from Mehri & Eskenazi (2020) – containing 60 dialogue contexts with human annotation on six model outputs. Based on the findings, the authors claim that ChatEval “delivers superior accuracy and correlation in alignment with human assessment.” In addition, they find that using diverse role prompts (using different personas) helps improve performance.

**Strengths:**

* Note that I’ve listed both strengths and weaknesses here as they are topically grouped together.
* The work provides a good list of related work and contextualizes itself in relation to prior/concurrent work. In doing so, however, it is unclear how the main goal of the proposed approach mirrors or differs from prior work (other than the proposed method itself; scoping the contribution/types of tasks more tightly can help here) and what kind of pros and cons or trade-offs there are in comparison to these other works that also leverage multiple LLMs. I’d encourage the authors to provide more information on this aspect both in introduction and related work, as I find the current explanation to be a bit hard to understand in the related work (i.e., “Concurrent with our work, Li et al. (2023b); Zhang et al. (2023) also propose a similar approach. However, they probe different dimensions of improving LLM-based evaluators and do not explore the effectiveness of natural language interaction.” In this sentence, what are “different dimensions” and what does “natural language interaction” mean here?)
* I think the idea of exploring diverse role specifications and communication strategies has a lot of potential. However, the current version that’s explored in the paper lacks depth and justification. For instance, there is one line justification for how the diverse role specification was formulated (“We take inspiration from Wu et al. (2023) and formulate an analogous role description.”), but it is unclear what other alternatives there are, why this is a good baseline to default to, what potential limitations are with this approach. Same thing applies to the design of communication strategies. The authors designed three strategies, but it is unclear how the design decisions were made and what factors are accounted for.
* Regarding the experiments, I have some concerns about the authors’ main claim. In Table 1, it’s unclear whether the difference between different methods is meaningful since it’s smaller than the difference between human annotators. More discussion on the inter-annotator agreement and how we should interpret the results would be necessary to make the claim about “superior accuracy and correlation”.  To account for a small number of instances, I’d highly encourage the authors to include standard deviations or errors on all tables and figures. On the other hand, I do appreciate that the authors delve into the importance of diverse role prompts, communication strategies, and the impact of role numbers and discussion turns. However, the qualitative analysis lacks nuance when reporting the finding; I’d be very cautious to call model behaviors “human-like” and “not just as a tool [...] but as an embodiment of interactive natural language dialogue” solely based on the four patterns observed in the experiments.
* Overall, I find the paper to have much room for improvement in terms of writing, need for more justification and explanation for design choices for the proposed method, and need more rigor in analyzing and reporting the experiment results.

**Weaknesses:**

Provided above.

**Questions:**

Minor suggestions
* In the abstract, I’d encourage the authors to include actual numerical results, as opposed to the textual description “superior accuracy and correlation”.
* In the abstract, it would be good to mention specific tasks or contexts that these claims are made.
* In Figure 1, if possible, showing actual (full) texts or abbreviated versions of texts (but longer than what it’s currently shown) would help readers understand how different model outputs (infused with different personas) contribute to overall performance boost.
* In the experiments section, the authors can clarify whether they use one model for all roles or use different models for different roles (it can be inferred from the results, but it’s better to clarify it before presenting the results).
* In Figure 4, the range on the y-axis should match in order to facilitate easy comparison between configurations.

---

> ### Author Response · Authors · 2023-11-17
> **Rebuttal Initial Response (1/2)**
>
> We express our genuine gratitude for the questions and concerns you've presented in your review. In response, we begin by re-emphasizing the main goal of our papers, which hold substantial significance. Subsequently, we address your specific concerns, offering explanations that we trust will illuminate the innovative aspects of our research and address the issues you've raised.
>
> **Main goal of our papers:**
> Most of the prior work utilizes a single LLM to score model responses. Recognizing the efficacy of single-LLM evaluations, we pondered how to transition from solitary evaluators to a coordinated system that integrates the decision-making of diverse evaluators, yielding more comprehensive and transparent conclusions. Given that impersonation (https://arxiv.org/abs/2305.14930) reveals LLMs strengths, we considered the MA approach through impersonative debating. We have also demonstrated that through linguistic communication, LLMs can effectively reach a consensus and provide superior evaluation outcomes.
>
> ---
> **Q1: Comparison with contemporary works**
>
> The contemporary research we referenced share the same aims to enhance the performance of llm-evaluators through multiple llms. However, by "different dimension," we intend to highlight that these works do not investigate the MA debate framework, and they overlook the potentiality and the heuristics of natural language communication—a primary focus of our paper. For instance, while the work (https://arxiv.org/abs/2307.02762) also employs multiple LLMs, their emphasis is on the discussion to takes into account each LLM's pairwise preferences of all answer pairs rather than the intricate components of debate frameworks, such as role design, communication strategy, moreover, their results have shown to be inferior to ours. The work (https://arxiv.org/abs/2308.01862), on the other hand, uses multiple LLMs to collate answers in an ensemble manner, aggregating results without the models engaging in a debate. We appreciate your feedback and will ensure that our distinctions from concurrent work are more explicitly stated in the revised version.
>
> ---
> **Q2: The origin of role design and communication strategy**
>
> Our role prompt design originated from existing literature as a foundational reference. We think that taking the existing design from the literature is a good start point for us to study the impersonation. As is shown in table 3, it also shows the effectiveness of the above mentioned role design. Additionally, we have also developed alternative designs and assessed their impact, with details available in section 4.1. We discovered that these specifically crafted role prompts perform better in their respective categories, as is shown in figure 3. Regarding communication strategy, the 'one-by-one' method establishes a common practice for turn-taking debates which is also adopted by previous framework (https://arxiv.org/abs/2303.17760). The two additional strategies we introduce are inspired by human interaction patterns and inherently have different functionalities in our framework. For example, 'simultaneous-talk' relaxes the rigid sequence of turns, while 'simultaneous-talk-with-summarizer' addresses the challenge of maintaining a lengthy conversation history. These functionalities are discussed in section 4.2. We are grateful for the reviewer's input and will further clarify these aspects in our forthcoming version.
>
> ---
>
> **Q3: Whether the difference between different methods is meaningful?**
>
> We have run the experiments for 5 times and included standard deviations in our results to showcase the consistent stability and statistically significant improvements rendered by our method. This speaks volumes about the robustness of our approach. Additionally, our objective is to demonstrate that our method can consistently outperform another as evidenced by the average and standard deviation. Therefore, if a method consistently yields results closer to the human average, we consider it to be superior.
>
> |                             |                        | Acc.        | Kap          |
> |------------------------------|------------------------|-------------|--------------|
> |ChatGPT                      | Single-Agent           | $53.7_{\pm 1.4}$ | $0.27_{\pm 0.02}$ |
> |ChatGPT                      | Multi-Agent (Ensemble) | $55.5_{\pm 0.7}$ | $0.29_{\pm 0.01}$ |
> |ChatGPT                     | Multi-Agent (ChatEval) | $60.0_{\pm 0.9}$ | $0.30_{\pm 0.02}$ |
> |GPT-4                         | Single-Agent           | $60.8_{\pm 0.7}$ | $0.36_{\pm 0.01}$ |
> |GPT-4                         | Multi-Agent (Ensemble) | $61.5_{\pm 0.5}$ | $0.38_{\pm 0.01}$ |
> |GPT-4                         | Multi-Agent (ChatEval) | $63.8_{\pm 0.9}$ | $0.40_{\pm 0.01}$ |

---

> ### Author Response · Authors · 2023-11-17
> **Rebuttal Initial Response (2/2)**
>
> **Q4: Further clarification of qualitative analysis**
>
> First of all, we appreciate the reviewers' recognition of our contributions through quantitative experiments.
> In terms of qualitative analysis, we would like to clarify that the term "human-like" used in our paper refers to the four patterns we've identified, which are heuristically derived from observing human behavior during debates. Noting the similarity in LLM debate processes, we initially labeled these as "human-like." However, considering your feedback, we acknowledge that the term might have been somewhat premature in the context of current AI capabilities. Consequently, we will amend the term to "human-inspired" to more accurately convey the essence of our observations.
>
> Regarding the concern about ChatEval as "not just a tool [...] but as an embodiment of interactive natural language dialogue," we want to emphasize that our debate framework transcends being merely an evaluative tool; it is instrumental in increasing the transparency and interpretability of AI systems. OpenAI has indeed articulated in their blog https://openai.com/blog/our-approach-to-alignment-research that debate is a technique intended to refine alignment research, aiding humans in evaluating complex tasks beyond direct human assessment. While their own work https://arxiv.org/abs/1805.00899 on debate has not extensively covered experiments in natural language and remains preliminary, our framework's utilization of natural language initiates a crucial dialogue on the evolving role of AI in evaluation processes. Our framework envisages AI as a collaborative partner in the evaluative discourse, not just the subject of evaluation.
>
> These discussions and the nuanced adjustments will be incorporated into the latest version of our paper. We believe that these revisions will not only address your concerns but also present our results more accurately, preserving the integrity of our original assertions about the dual contributions of our method in enhancing both quantitative performance and providing qualitative insights.
>
> ---
>
> **Q5: Minor suggestions**
>
> Thank you for your advice for improving our paper, we will take these issues into account and incorporate the necessary amendments in the forthcoming version of our paper.

---

> > ### Comment · Reviewer_XZLb · 2023-11-22
> >
> > I thank the authors for their response and have adjusted the score.

---

> > > ### Author Response · Authors · 2023-11-23
> > >
> > > Dear reviewer XZLb:
> > >
> > > We have observed the positive adjustment in our paper's evaluation, and we are truly thankful for your acknowledgment of its potential. We trust that our initial responses have addressed most of your concerns, and we are more than welcome for any further comments you may have.
> > >
> > > Once more, we wish to convey our deep appreciation for the time and effort you have dedicated to reviewing our work.

---

> ### Author Response · Authors · 2023-11-21
>
> Dear reviewer:
>
> We are deeply appreciative of your insightful and constructive feedback, which has played a crucial role in elevating the quality of our manuscript.
>
> In response to your comments, we have diligently worked to address the raised concerns and have supplemented our paper with additional results to further solidify its contributions.
>
> Please feel free to reach out if you require further clarifications or suggest any additional experiments that could more comprehensively assess our work.
>
> Should our revisions and detailed responses meet your initial concerns, we would be immensely grateful for your consideration in positively adjusting the evaluation of our paper. Your support is of great importance to us.
>
> Once again, we thank you for your dedicated time and effort in reviewing our submission.

---

### Official Review · Reviewer_rxGE · 2023-11-01

**Soundness:** 3 good
**Presentation:** 3 good
**Contribution:** 3 good
**Rating:** 6
**Confidence:** 4

**Summary:**

This paper proposes ChatEval, a multi-agent framework for text evaluation that simulates human collaborative and argumentative dialogue.
ChatEval demonstrates superior performance on two benchmarks, FairEval and Topical-Chat, compared to single-agent and other LLM-based methods.

**Strengths:**

1. The work has implications for the field of text evaluation, which aligns better with human preferences.

2. The authors have conducted extensive experiments and provided a thorough analysis of the results. The proposed method outperforms single-agent and other LLM-based methods on two benchmarks, demonstrating its effectiveness.

3. The work also highlighted the importance of diverse role prompts, which is a valuable insight for future research.

**Weaknesses:**

1. Figure 4 does not show whether simultaneous-talk-with-summarizer can outperform one-by-one. Although the chart has an upward trend, it is still necessary to further increase the Role Numbers and Discussion Turns to prove the author's point of view.

2. The specific examples of three different communication strategies are lacking, which can be put into the appendix.

3. The resource cost of ChatGPT/GPT-4 is a problem that needs to be considered, and the paper does not compare the resource consumption of previous methods with the proposed method.

**Questions:**

1. About fairness comparison. The setting of ChatEval is 2 agents with 2 discussion turns. It is not clear what the setting of Multi-Agent (Ensemble) is like.

2. More detailed about the Ensemble method.

---

> ### Author Response · Authors · 2023-11-17
>
> Your constructive comments are highly appreciated. We have detailed our responses to your insightful questions and concerns below:
>
> ---
>
> **Q1：Whether simultaneous-talk-with-summarizer outperform one-by-one**
>
> As is shown in Figure 4,  'simultaneous-talk-with-summarizer' communication strategy exhibits an upward trend, achieving higher accuracy than the 'one-by-one' method under certain conditions, such as increased role numbers. However, our intention of the analysis is not to rank these strategies but rather to highlight the potential of a component that has been previously overlooked in literature. Prior studies on multi-agent debate have predominantly focused on the 'one-by-one' strategy, which we believe leaves room for further enhancement by considering various communication strategies.
>
> ---
>
> **Q2: Specific examples of different communication strategy**
>
> Currently, the comprehensive algorithm is detailed in the appendix, and we will include illustrative examples in the appendix of our next version to facilitate a better understanding.
>
> ---
>
> **Q3: Time and money cost**
>
> We have assessed the resource expenditure, and the following table shows our method can substantially lower both time and labor costs in comparison to the employment of human annotators.
>
> |                | Cost    | Time   |
> |-------------------|---------|--------|
> |Human             | \$90    | 240min |
> |gpt-4 (SA)    | \$2.90  | 10min  |
> |gpt-4 (MA ensemble)  | \$8.70  | 10min  |
> |gpt-4 (MA chateval)  | \$12.30 | 36min  |
> |Chatgpt(SA)   | \$0.11  | 3min   |
> |Chatgpt(MA ensemble) | \$0.33  | 3min   |
> |Chatgpt(MA chateval) | \$0.41  | 11min  |
>
> ---
>
> **Q4: Details about ensemble method**
>
> To clarify the use of the ensemble method in our paper, the term 'ensemble' as used in our context refers to the procedure where we apply the SA Chain of Thought (COT) method multiple times using different role prompts across the same instance and then aggregate their results through averaging to derive the final outcome. We will amend our paper to explicitly define this process and ensure that the methodology is understandable for all readers.

---

> > ### Comment · Reviewer_rxGE · 2023-11-22
> >
> > Thank you for your response. Is the cost of baseline methods like G-EVAL available?

---

> > > ### Author Response · Authors · 2023-11-22
> > > **Response to the Follow-Up Question**
> > >
> > > Dear reviewer:
> > >
> > > We express our heartfelt thanks for your prompt response. Your diligent efforts in reviewing our work are highly respected and we sincerely appreciate the affirmative evaluation you've provided for our paper thus far. In accordance with the cost analysis of these baseline methods, we have included an additional table on the FairEval benchmark for your reference.
> > >
> > > It is important to highlight that the mentioned methods require multiple inferences per instance, which we have assumed to be run in an asynchronous manner. Consequently, their time cost would be that of an SA COT method, consistent with our reporting for our ensemble methods previously discussed.
> > >
> > > Furthermore, regarding the g-eval method, which necessitates sampling 20 times to estimate the token probabilities, the incurred cost is literally 20-fold that of an SA COT method.
> > >
> > > We hope that these additional results address your concerns. If our recent revisions and clarifications meet your expectations, we would be grateful if you would consider supporting the paper by further increasing the score.
> > >
> > >
> > > |                | Cost    | Time   |
> > > |-------------------|---------|--------|
> > > |G-eval-3.5 | \$2.2   | 3min |
> > > |G-eval-4    | \$58  | 10min  |
> > > |FairEval-chatgpt | \$0.34  | 3min  |
> > > |FairEval-gpt4  | \$6.38 | 10min  |

---

### Official Review · Reviewer_zL85 · 2023-11-02

**Soundness:** 2 fair
**Presentation:** 2 fair
**Contribution:** 3 good
**Rating:** 5
**Confidence:** 4

**Summary:**

The paper introduces a new way of using LLMs for text evaluation through multi-agent discuss and evaluation. Instead of using the LLM to generate one round of evaluation, the authors propose to design multiple LLM agents with diverse roles, and let them go through several rounds of discussion to make the final evaluation. Results on one open question answering dataset, and one dialogue dataset demonstrates better alignment with human judgement for the proposed multi-agent framework. The authors also conduct additional experiments to examine individual components of the system, including the design of communication pattern among agents, diverse role of agents, number of agents involved and number of discussion rounds, which are helpful for future study.

**Strengths:**

1. The authors carefully study the different components of the multi-agent system, including the communication pattern, agent role design, number of agents and discussion rounds. The additional experiments for individual components and detailed discussion are very helpful for understanding the design choices and inform future research.
2. Results on two benchmarks showcase improvement of using multi-agent with both ChatGPT and GPT-4.

**Weaknesses:**

1. The experiments are conducted only on two datasets that are relative small. On Topical-Chat the win/loss of single-agent and multi-agent is mixed on different rubrics, and ChatGPT and GPT-4 seem to have different behaviors. Some more analysis on each rubric will be helpful rather than simply compare the improvement on average.
2. The experiments are conducted only with ChatGPT and GPT-4, thus unclear whether the proposed method could generalize to other LLMs, or limiting to GPT models. Evaluation on more models of different capacities could help understand when the multi-agent framework would work, is there emergent ability that require model with decent performance to have constructive discussion and evaluation.
3. There is no discussion on limitation of the proposed method. For example, the inference speed/cost will get affected given that multi-agent requires more rounds of generation with the LLM.
4. Some implementation details could be clarified.
    1. In section 3.2, it is mentioned that the default results are obtained with 2 agents, what are these 2 agents? Appendix describes the implementation of 5 agents, while in section 4.1 it is mentioned that there are three different roles.
    2. In section 4.1, the authors mentioned comparison between specific roles and default setting, but seems Figure 3 only includes results of specific roles.
    3. What is the specific version of ChatGPT and GPT-4 used in the paper, since that will affect model performance as well.

**Questions:**

Please see weakness above. In particular the clarification on implementation details/experiments and discussion on the limitation.

**Details Of Ethics Concerns:**

It would be good to have some discussion on ethics consideration on using LLMs for evaluation, in particular how it works together with human evaluation.

---

> ### Author Response · Authors · 2023-11-17
>
> Thank you for your comprehensive feedback. We have carefully considered your points and offer our detailed responses to each concern and question below:
>
> ---
>
> **Q1: Analysis of different sub-metrics**
>
> Our findings indicate that on average, our MA approach surpasses the SA method in performance when using both ChatGPT and GPT-4. However, there are subtle variations in performance across different sub-metrics between the two LLMs. Table 2 shows that for ChatGPT, MA slightly underperforms in Naturalness and Coherence but shows improvements in Engagingness and Groundedness. In contrast, MA of GPT-4 displays a more uniform improvement across three sub-metrics, with only a minor decline in Groundedness. It is also important to highlight that such variability is observed not only within our proposed metrics but also within traditional metrics and previous methods like g-eval. For instance, g-eval-4's performance dips in Engagingness and Groundedness compared to g-eval-3.5. It is notable that our GPT-4 (ChatEval) consistently excels over ChatGPT (ChatEval) in both SA and MA configurations, which is a trend we consider to be a reasonable and scalable behavior.
>
> ---
>
> **Q2：Questions on the LLMs we choose**
>
> Our focus on ChatGPT and GPT-4 stems from their recognition in prior studies as benchmarks for evaluating LLM-based evaluators https://arxiv.org/abs/2303.16634. They are the mostly widely used llm-based evaluators in various scenerio and has been shown effective under SA settings. This established efficacy provides a solid foundation for our decision to build upon ChatGPT and GPT-4 for our MA framework. It is reasonable to extend their proven capabilities in a SA context to a more complex MA setting, which can potentially harness and amplify the strengths of these models. In comparison, some other smaller models diverge from human preference even in SA setting, let along the multi-agent debate framework. To ensure consistency and comparability with previous research, we believe it is appropriate to conduct our experiments with these two models.
>
> ---
>
> **Q3: Time and money cost**
> Here we include the cost analysis table. The results indicate that while the MA debate inherently incurs higher costs, it remains significantly more cost-effective than using human annotators, both in terms of time and expense. We will incorporate this cost analysis into the forthcoming version of our manuscript.
>
> |                 | Cost    | Time   |
> |-------------------|---------|--------|
> |Human             | \$90    | 240min |
> |gpt-4 (SA)    | \$2.90  | 10min  |
> |gpt-4 (MA ensemble)  | \$8.70  | 10min  |
> |gpt-4 (MA chateval)  | \$12.30 | 36min  |
> |Chatgpt(SA)   | \$0.11  | 3min   |
> |Chatgpt(MA ensemble) | \$0.33  | 3min   |
> |Chatgpt(MA chateval) | \$0.41  | 11min  |
>
> ---
>
> **Q4: Implementation details**
>
> 1. In the experiments involving two agents, we utilize the roles of 'general public' and 'critic'.
> 2. In Figure 3, we have compared specific roles against the default setting. Here, 'general' on the x-axis refers to the default setting.
> 3. Regarding the models referenced, "ChatGPT" corresponds to gpt-3.5turbo-0301 and "GPT4" to gpt-4-0314.

---

> > ### Comment · Reviewer_zL85 · 2023-11-20
> > **Follow up on author response**
> >
> > Thanks for the detailed response. Some follow up on question 2 about the selection of LLMs. I agree that ChatGPT and GPT4 are widely adopted as evaluators in the literature thus should definitely be included. But I still feel that since the focus is the multi-agent frame work with LLMs, it is still important to examine other LLMs and not GPT models alone.
> > Meanwhile, seems there is an assumption is that the LLM must first be good at single-agent evaluation setting in order to succeed in the multi-agent scenario, is this true. Actually I feel if through multi-agent debate weaker LLMs can obtain improvement and get close to GPT performance, it is also valuable, and might be a great choice to reduce cost as well.

---

> ### Author Response · Authors · 2023-11-21
> **Response to follow up questions**
>
> We have now included additional results for Llama2-chat-7b and Vicuna-7b-v1.5 and attached these new findings for your review. As indicated in the updated table, Llama2 achieves only marginal performance under single-agent CoT methods. Despite reaching an accuracy of 37.5%, the negative kappa coefficient suggests that the observed agreement is less than what would be expected by chance. However, it's noteworthy that through the application of a multi-agent debate, Llama2 demonstrates a modest improvement, albeit still marginally above chance levels.
>
> In contrast, Vicuna exhibits more robust performance improvement compared to Llama2. ChatEval notably enhances its capabilities, achieving an accuracy of 52.3% and a kappa coefficient of 0.19, indicative of fair agreement beyond chance. While both models show limitations in their performance, **these results demonstrate the effectiveness of ChatEval and underscore its positive impact**.
>
> We are grateful for the reviewers' insights regarding the application of ChatEval to less powerful models. Our findings demonstrate that ChatEval can still significantly enhance performance, a point we believe holds substantial value, as also echoed by the reviewers.
>
> Furthermore, we hope that our revisions and comprehensive responses have satisfactorily addressed the initial concerns. If so, we would be deeply appreciative of your consideration in revising your evaluation of our paper upwards.
>
> Thank you once again for your invaluable feedback and guidance in refining our study.
>
>
> |                             | Acc. | Kap.  |
> |------------------------------|------|-------|
> |Llama2-chat-7b (Single)   | $37.5_{\pm 1.3}$ | $-0.01_{\pm 0.01}$ |
> |Llama2-chat-7b (ChatEval) | $40.0_{\pm 1.1}$   |  $0.05_{\pm 0.01}$  |
> |Vicuna-7b-v1.5 (Single)       | $45.0_{\pm 1.5}$ |  $0.12_{\pm 0.01}$  |
> |Vicuna-7b-v1.5 (ChatEval)     | $52.3_{\pm 1.3}$ | $0.19_{\pm 0.01}$  |

---

### Official Review · Reviewer_asDf · 2023-11-08

**Soundness:** 3 good
**Presentation:** 3 good
**Contribution:** 3 good
**Rating:** 6
**Confidence:** 4

**Summary:**

In this work, the authors propose a new framework called ChatEval that contains multiple debater agents to evaluate the quality of the text generated.  Each debater agent is an LLM aimed to generate responses based on prompts and there are 3 communication strategies followed between the multiple debater agents.
The authors evaluate the proposed approach on two benchmarks Fair Eval and Topical Chat and find that the proposed framework aligns closely with the human preferences.

**Strengths:**

1. The paper addresses an important problem of evaluating textual generations by using LLMs and trying to reduce the shortcomings faced with traditional human-based evaluations.
2. Results demonstrate the framework's effectiveness on the FairEval and Topical Chat dataset by comparing it against various other frameworks on this dataset.

**Weaknesses:**

1. In the abstract, the authors mention that one of the drawbacks of using humans in the pipeline is the time and labor cost. The analysis of the proposed framework would benefit significantly if there is any analysis in terms of time spent by humans for annotation vs LLM.
2. The framework seems to be suited only for short conversations based on the Analysis in Section 4.3. This would be an issue when extending this framework for evaluating longer dialogue conversations.

**Questions:**

1. In Tables 1 and 2 why is only Multi Agent (ChatEval) using the strategy of One-by-one mentioned? Why are results from other strategies not provided in the table?
2. In Section 3.3, it is claimed that Chateval surpasses Fair Eval's best results but this does not seem to be the case with regards to the Kappa correlation coefficient.
3. How effective is the framework at detecting issues on hallucination which is quite common along LLMs?

---

> ### Author Response · Authors · 2023-11-17
>
> We highly value your detailed feedback on our study. Please find below our response to the key issues highlighted in your comments.
>
> ---
>
> **Q1: Time and money cost analysis**
>
> We extend our gratitude to the reviewer for highlighting the importance of a time cost analysis in supporting our proposed framework. In the appended table, it is evident that the evaluator based on LLMs significantly reduces both time and financial expenditures. Although the chateval framework incurs higher costs than the SA  and MA ensemble methods due to multiple inference rounds necessary for final judgment derivation, it remains more cost-effective than employing human evaluators. We consider this an acceptable trade-off for the benefits provided.
>
>  |                | Cost    | Time   |
> |-------------------|---------|--------|
> |Human             | \$90    | 240min |
> |gpt-4 (SA)    | \$2.90  | 10min  |
> |gpt-4 (MA ensemble)  | \$8.70  | 10min  |
> |gpt-4 (MA chateval)  | \$12.30 | 36min  |
> |Chatgpt(SA)   | \$0.11  | 3min   |
> |Chatgpt(MA ensemble) | \$0.33  | 3min   |
> |Chatgpt(MA chateval) | \$0.41  | 11min  |
>
> ---
>
> **Q2: Challenges of long context issue**
>
> We acknowledge the reviewer's point regarding the challenges of evaluating prolonged conversations, a concern shared by both academic and industrial spheres.  Within our framework, we have incorporated a summary-style communication strategy specifically designed to mitigate the issues presented by lengthy historical context. Specifically, at the end of each iteration of the debate, we will prompt an extra LLM to summarize the messages conveyed so far and use this condensed information as its historical context. The results in section 4.3 show that summary-style has a better scalable behaviour. Additionally, we would like to emphasize that the principal aim of our work is to establish a more comprehensive mechanism for evaluating textual content. The experimental results indicate that the application of a debating format can indeed catalyze improved performance. As more capable LLMs emerge, particularly those adept at managing exceptionally long contexts, our framework is inherently equipped to generalize these advancements. The integration of such LLMs is anticipated to unlock the full potential of our framework further.
>
> ---
>
> **Q3: Additional results for other communication strategy**
>
> Initially, we highlighted the 'one-by-one' strategy for ChatEval as it was our primary focus and the most direct method to benchmark against existing frameworks. We agree that it is important to show a comprehensive view of the performance across different strategies. To this end, we have now included results from additional strategies in the revised tables. The data presented in the table indicate that each of the strategies yields a statistically consistent enhancement over the SA method, affirming the efficacy of our approach. For an in-depth examination of communication strategies, we recommend that reviewers refer to Section 4.2.
>
>
> |                             |                       | Acc.        | Kap          |
> |------------------------------|------------------------|-------------|--------------|
> |ChatGPT                      | SA         | $53.7_{\pm 1.4}$ | $0.27_{\pm 0.02}$ |
> |ChatGPT                      | one-by-one           | $60.0_{\pm 0.9}$ | $0.30_{\pm 0.02}$ |
> |ChatGPT                      | simultaneous-talk | $59.3_{\pm 1.0}$ | $0.29_{\pm 0.02}$ |
> |ChatGPT                     | simultaneous-talk-with-summarizer | $58.8_{\pm 1.3}$ | $0.28_{\pm 0.03}$ |
> |GPT-4                         | SA         | $60.8_{\pm 0.7}$ | $0.36_{\pm 0.01}$ |
> |GPT-4                         | one-by-one           | $63.8_{\pm 0.9}$ | $0.40_{\pm 0.01}$ |
> |GPT-4                         | simultaneous-talk | $63.2_{\pm 0.6}$ | $0.38_{\pm 0.02}$ |
> |GPT-4                         | simultaneous-talk-with-summarizer | $63.4_{\pm 0.8}$ | $0.39_{\pm 0.02}$ |
>
>
> ---
>
> **Q4: Reviewer's question on Chateval underperform Faireval**
>
> Upon reviewing the results presented in our paper, we kindly request the reviewer to re-examine the figures in table 1. It appears that the performance of chateval exceeds that of faireval, particularly concerning the Kappa correlation coefficient.
>
> ---
>
> **Q5: Hallucination issues**
>
> The issue of hallucination in LLMs is indeed prevalent, and past research has suggested that multi-agent debates can mitigate this to a certain extent https://arxiv.org/abs/2305.14325. However, we consider a thorough investigation into this matter to be beyond the scope of the current paper. We believe that integrating external resources, such as search engines or databases, could be an effective strategy to curtail hallucinations. We plan to explore these possibilities and assess their compatibility with our debate framework in subsequent research endeavors.

---

> ### Author Response · Authors · 2023-11-23
>
> Dear reviewer asDf:
>
> We extend our sincere thanks for the dedication and attention you've invested in the review process of our manuscript for ICLR. Your commitment, particularly in these demanding times, is deeply valued.
>
> We are also grateful for the positive score you've assigned, which acknowledges the potential of our work. We have made every effort to address the concerns raised by the reviewers. In the rebuttal period, we have provided additional results and clarifications that significantly enhance the quality of our manuscript and we are also eager to learn whether our responses have adequately addressed your concerns. If you believe that our feedback has resolved the issues you raised, we would greatly appreciate your continued support for our submission.
>
> Once again, we are thankful for your constructive feedback and expert guidance.

---

### Official Review · Reviewer_2Bwn · 2023-11-09

**Soundness:** 3 good
**Presentation:** 3 good
**Contribution:** 3 good
**Rating:** 6
**Confidence:** 4

**Summary:**

This paper focuses on the task of automatic text evaluation and highlights the limitations of existing n-gram metrics in correlating with human judgments. The authors propose ChatEval, a framework leveraging LLMs for automatic text evaluation. Instead of prompting a single LLM to assess the generation quality, ChatEval integrates multiple LLMs, facilitating debates among them to enhance the robustness and human-like quality of evaluation results.

**Strengths:**

1. The idea of using LLMs for automatic text evaluation is intriguing and holds potential.
2. The paper is well-structured and easy to follow. The inclusion of informative figures and tables enhances clarity.

**Weaknesses:**

1. The paper may benefit from providing more in-depth technical details about the ChatEval framework. While a prompt template is present in the appendix, the determination of LLM roles and the potential impact of varying roles and orders remain unclear. In addition, the framework relies on prompting LLMs, however, the paper lacks sufficient information on the design of prompts and their robustness.
2. The evaluation results present a challenge in assessing whether the multi-agent debate framework outperforms existing LLM-based evaluators in general cases. Addressing questions related to the listed questions would contribute to a more informed judgment.

**Questions:**

1. As mentioned in section 3.2, two agents are employed in the implementation of ChatEval. Can they always reach an agreement at the end of debates?

2. Regarding Table 2, does "MA" in the table represent multiple agents employed in the debating way. Are there corresponding results for multiple LLMs used in an ensemble manner, aligning with the methods in Table 1?

3. Considering the performance gap between G-EVAL-3.5 and ChatGPT(MA) in Table 2 for dialogue response generation, it appears that the effectiveness of the proposed framework is influenced by the chosen LLM. Has there been an evaluation of ChatEval's generalization ability?

4. It seems that the G-Eval framework can be easily adapted to open-ended QA evaluation. From the performance differences in Table 2 between G-Eval and ChatEval, I wonder whether the multiple-agent debating framework really works better than the single-agent COT framework?

---

> ### Author Response · Authors · 2023-11-17
>
> We are grateful for your thoughtful observations on our research approach. In this response, we aim to address the points you've raised.
>
> ---
>
> **Q1: The origin and the impact of varying role design as well as order issue**
>
> We choose the role design introduced in https://arxiv.org/abs/2303.15078 as a starting point to specify role prompts and facilitate a fair comparison with prior literature. As for roles design alternatives, we believe that the experiments presented in Section 4.1 probe the impact of different roles design. We design roles for different groups, and these roles perform better in their respective categories. As evident in the results, we believe the impact of role design is significant. In terms of the order issue, we have proposed "simultaneous-talk" communication strategy which represents a pragmatic solution to alleviate strict order constraints in contrast to one-by-one communication. It employs asynchronous functions to merge historical messages, shuffling the order of messages within the history at each round. This approach enables a flexible and dynamic interactions among agents. We will make it clearer in our next version of the paper.
>
> ---
>
> **Q2: Inter-annotator agreement among agents**
>
> In response to your question about agreement, we provide an analysis of the inter-annotator agreement experiment results in the following table. We employed ICC (Intraclass Correlation Coefficient) to measure the agreement among annotators. From the table, it is evident that the annotators are generally able to reach a consensus by the end of their debate. We also observe that as we increase the number of roles, there is a decrease in agreement. It is reasonable considering that it's not also easy for a team of people to reach a consensus. Still, the overall ICC values remain at a high level, indicating strong agreement among annotators.
>
>
>   |     | 2 roles | 3 roles | 4 roles | 5 roles |
> |-----|---------|---------|---------|---------|
> | ICC | 0.981   | 0.955   | 0.890   | 0.881   |
>
>
> ---
>
> **Q3: Ensemble results in Table 2**
>
> Here are the results for the ensemble method applied to Topical-Chat benchmark. It is evident that the ensemble method in Topical-Chat shows some improvement compared to SA. Nevertheless, it still falls behind our Chateval framework, highlighting the effectiveness of our proposed framework.
>
> |                   | Nat| | Coh|       | Eng|       | Grd|       | Avg|       |
> |-------------------|---------|-------|-------|-------|-------|-------|-------|-------|-------|-------|
> |                   |  $\tau$ |   $\rho$    |  $\tau$     |   $\rho$    |   $\tau$    |   $\rho$    |   $\tau$    |   $\rho$    |   $\tau$    |     $\rho$  |
> | ChatGPT(SA)       | 0.474   | 0.421 | 0.527 | 0.482 | 0.599 | 0.549 | 0.576 | 0.558 | 0.544 | 0.503 |
> | ChatGPT(Ensemble) | 0.421   | 0.359 | 0.486 | 0.442 | 0.611 | 0.551 | 0.661 | 0.628 | 0.545 | 0.495 |
> | ChatGPT(ChatEval)       | 0.441   | 0.396 | 0.500 | 0.454 | 0.664 | 0.607 | 0.602 | 0.583 | 0.552 | 0.510 |
> | GPT-4(SA)         | 0.532   | 0.483 | 0.591 | 0.535 | 0.734 | 0.676 | 0.774 | 0.750 | 0.658 | 0.611 |
> | GPT-4(Ensemble)   | 0.512   | 0.450 | 0.607 | 0.544 | 0.755 | 0.693 | 0.781 | 0.756 | 0.664 | 0.611 |
> | GPT-4(ChatEval)         | 0.630   | 0.571 | 0.619 | 0.561 | 0.765 | 0.695 | 0.722 | 0.700 | 0.684 | 0.632 |
>
> ---
>
> **Q4: Generalization of ChatEval and its performance compared to G-eval**
>
> We would like to clarify that our method is also applicable to a range of LLMs and it operates orthogonally to SA prompting techniques. It complements and extends these techniques by integrating them within a MA debate context, which can leverage the unique strengths of different LLMs in a collaborative setting. Looking ahead, we are excited about the potential of more specialized and expert-driven LLMs in different domains, which promise to further generalize our framework.
>
>  It is important to note that while we benchmarked our approach against the g-eval method, the comparison may not be entirely straightforward. The g-eval method, which achieved a better 3.5 performance score, does not solely rely on COT prompting; instead, it requires access to the language model's logits. These logits are then used to perform a weighted calculation, which differs from our methodology. However, our proposed MA approach does not necessitate access to the model's logits—a requirement that is often impractical in real-world applications. Consequently, it is expected that there would be performance discrepancies between our method and g-eval.
>
> Additionally, our experimental results demonstrate that the MA approach (MA ChatEval), surpasses the performance of the SA COT method referenced in our main text. This responds to the reviewers' queries by confirming that our MA approach surpasses the SA CoT method, as clearly demonstrated by our experimental results.

---

> > ### Comment · Reviewer_2Bwn · 2023-11-30
> >
> > Thank you for providing additional explanations and presenting extra results, which have contributed to addressing some of my concerns. I have revised the scores accordingly. While I acknowledge the potential of using LLMs for automatic evaluation and the efficiency of multiple LLMs working together, my concern lies in the paper's claim of various communication strategies and diverse role prompts as contributions. I expected more a more thorough investigation into the assignment of roles and the design of an effective communication framework. Though there is brief analysis regarding the comparison of different strategies, the current version seems somewhat heuristic, lacking a clear guideline for the configuration of multi-agent debates.

---

> ### Author Response · Authors · 2023-11-23
>
> Dear reviewer 2Bwn:
>
> We express our sincere gratitude for the time and effort you have invested in reviewing our manuscript, and for the valuable insights provided. We have dedicated considerable effort to address the concerns initially raised, including **Inter-annotator agreement** , **Ensemble results**, and we would like to also highlight that we have also addressed concerns raised by other reviewers, including the results on weaker models illustrating the **Generalization** of ChatEval.
>
> These collective responses, we hope, demonstrate our commitment to advancing the quality of our work. If you find that our revised manuscript meets your professional expectations, we would be deeply appreciative of your consideration for further supporting the paper.
>
> Once again for your expert guidance and recognition of our work.

---

### Meta-Review · Area_Chair_GYkq · 2023-12-05

**Metareview:**

This paper introduces ChatEval, a novel multi-agent debate framework that utilizes multiple LLMs to evaluate the quality of machine-generated texts. It tries to address the limitations of existing n-gram metrics in text evaluation by proposing a collaborative approach in which multiple LLMs engage in debates to enhance evaluation robustness and human-like quality. In contrast to prior work, ChatEval employs multiple debater agents with diverse roles and communication strategies, thereby improving alignment with human judgment. Additional contributions include outperforming single-agent and other LLM-based methods on benchmarks such as FairEval and Topical-Chat. While most reviewers acknowledged the paper's contributions, they also expressed various concerns about a relative lack of analyses, such as investigations into the assignment of roles and the design of an effective communication framework. Some aspects of the work seem also heuristic. The reviewers also requested experiments with different LLMs and analyses of time and money costs. The authors addressed these requests during the author-reviewer discussion. Given that most reviewers are satisfied with the paper's level of contribution, and that the author responses leave no major concerns, I recommend accepting this paper. Given the reviewers' various requests for more analyses, I urge the authors to address their requests in the camera-ready paper.

**Justification For Why Not Higher Score:**

Some aspects of the work appear somewhat heuristic, particularly in lacking a clear guideline for the configuration of multi-agent debates.
The experiments are conducted on two datasets that are relatively small.

**Justification For Why Not Lower Score:**

Significant contributions: introduction of a novel multi-agent framework for text generation evaluation; the proposed method outperforms single-agent and other LLM-based methods on two benchmark.

---

### Decision · Program_Chairs · 2024-01-16

Accept (poster)